# SEAT: Sparsity Enhanced Attention in the frequency domain for Time Series forecasting

## Abstract

Transformer-based models have demonstrated superior performance in multivariate time series forecasting, leveraging attention mechanisms. However, existing transformer-based methods tend to overlook the high similarity among adjacent time steps and the correlation between multivariate time series. This often results in *block-wise* attention patterns that hinder efficient global information capture, thereby limiting the model's representation capacity and degrading prediction accuracy. In this work, we mathematically characterize and theoretically validate this limitation, showing how it undermines the stability of learned representations and restricts effective feature extraction. To alleviate this issue, we propose a lightweight and model-agnostic framework named *Sparsity Enhanced Attention in the Frequency Domain* (**SEAT**). By projecting time series data into the frequency domain, SEAT reconstructs the attention matrix to mitigate block-wise patterns, thereby enhancing the model's ability to capture global temporal dependencies. As a plug-and-play module, SEAT can be integrated into existing transformer-based architectures without altering their core structure. Extensive experiments on standard benchmarks demonstrate that SEAT consistently improves the predictive performance while preserving computational efficiency.

## 1 Introduction

Transformer-based models have achieved remarkable success in multivariate time series forecasting (Demirel et al., 2012; Patton, 2013; Su et al., 2025; Huang et al., 2024; Liu et al., 2024b; Li et al., 2019; Wu et al., 2020), largely due to their capacity to capture long-range temporal dependencies through self-attention mechanisms (Su et al., 2025; Huang et al., 2024; Liu et al., 2024b; Li et al., 2019; Wu et al., 2020). However, directly applying attention mechanisms—originally developed for NLP or vision (Vaswani et al., 2017; Dosovitskiy et al., 2020)—to time series data can be suboptimal, as these mechanisms often fail to account for the inherent characteristics of raw time series data—particularly, the smoothness and high similarity across adjacent time steps. This local smoothness can lead to attention mechanisms to assign redundant weights, thereby weakening representation abilities.

To improve forecasting performance, recent works introduced novel architectures to enhance temporal and variable-level modeling (Nie et al., 2023; Chen et al., 2024; Wu et al., 2021; Liu et al., 2022a). While effective, these methods are often model-specific and require substantial redesign. This raises a complementary question: rather than adapting models to the data, *can we instead construct a model-agnostic representation that aligns more naturally with the inductive biases of attention mechanisms, enabling standard architectures to perform better without structural changes?*

Addressing this question requires a deeper understanding of time series characteristics. A relevant observation was previously made in Crossformer (Zhang & Yan, 2023), which reported a segmental pattern in attention distributions during long-term forecasting: temporally adjacent tokens tend to receive nearly identical attention scores. Building upon this insight, we conduct a comprehensive analysis across a suite of Transformer-based models and identify a recurring structural pattern in attention matrices—namely, the emergence of *block-wise attention*, where rectangular regions of similar intensity dominate the attention maps (Figure 1b). Despite its prevalence, the root causes and consequences of this structure remain largely unaddressed. We formalize this observation within a

theoretical framework, providing a mathematical account of how redundancy in attention weights arising from signal smoothness affects representational capacity.

In this work, we propose *Sparsity-Enhanced Attention in the Frequency Domain* (SEAT), a lightweight and plug-and-play module that strengthens the expressive capacity of Transformers for multivariate time-series forecasting. Our design builds on two key insights: (1) time-series signals are inherently sparse in the frequency domain, and (2) the entanglement of temporal smoothness and cross-variable correlations drives the block-wise collapse of attention. To address this, SEAT first transforms each variable's sequence into the frequency domain, where spectral preconditioning disentangles local smoothness from cross-variable redundancy and raises the effective rank of attention. A learnable frequency-gated enhancement then reintroduces discriminative structure before projecting back to the time domain. This process enriches global modeling capacity and substantially alleviates block-wise attention degeneration (Figure 1c).

Our contributions can be summarized as follows:

- **Theoretical foundation.** We characterize *block-wise attention* in Transformer-based forecasting, showing how auto-correlation and cross-variable dependence cause low-rank structure, entropy loss, and gradient alignment. We also show that spectral preconditioning and near-isometric projection act as whitening and rank-preserving steps, yielding testable predictions on entropy and rank.

- **Method and empirical evidence.** We introduce SEAT, a frequency-domain framework that leverages signal sparsity and disentanglement to enhance temporal modeling. The module is lightweight, plug-and-play, and improves the effective rank of attention representations.

- **Empirical validation.** On eight real-world datasets demonstrate that our module consistently improves forecasting accuracy and achieves state-of-the-art performance, while maintaining efficiency and broad compatibility with Transformer-based architectures.

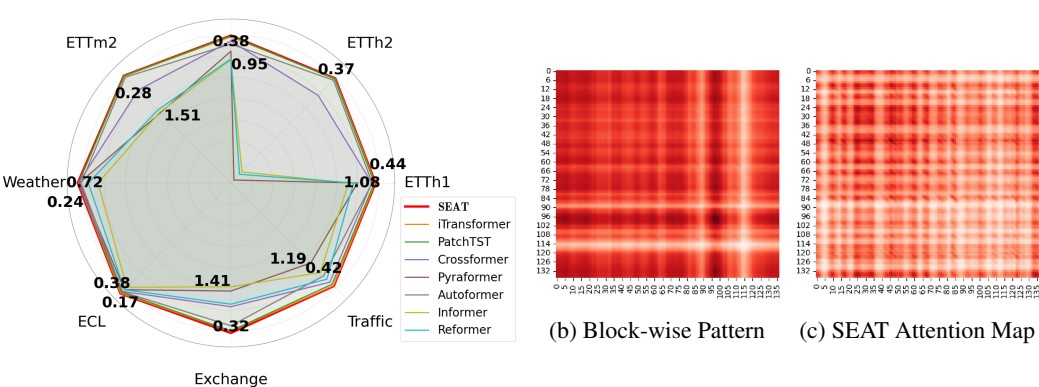

(a) Performance Comparison

(b) Block-wise Pattern

(c) SEAT Attention Map

Figure 1: (a) Radar plot of MSE across eight datasets, where lower values denote better performance. SEAT (red line) consistently outperforms or matches baseline models. (b) Transformer attention heatmap showing block-wise patterns, with adjacent time steps receiving similar attention weights. (c) SEAT attention heatmap exhibiting greater distinction, suggesting stronger representational ability.

## 2 PRELIMINARIES

**Problem setting.** We study multivariate time series forecasting with input windows $x_{1:L} \in \mathbb{R}^{L \times d}$ (rows are time), where Transformer-style attention is applied along the temporal axis. Empirically, mainstream models (e.g., iTransformer, Crossformer, PatchTST, Informer; see Figure 3) often produce *block-wise* attention maps on long sequences: each query attends heavily to a small, contiguous neighborhood in time, producing near-duplicate attention rows (local "blocks"). This limits representation diversity, harms long-range dependency modeling, and makes optimization vulnerable to gradient alignment.

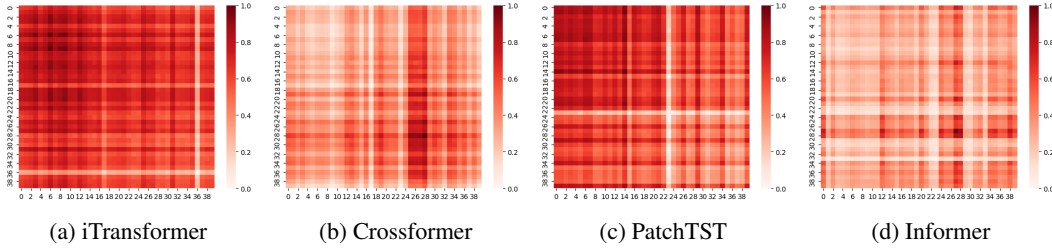

|        (a) iTransformer        |        (b) Crossformer        |        (c) PatchTST        |        (d) Informer        |

Figure 3: Representative attention heatmaps on ETTm2 (more in Appendix F). Local "blocks" indicate redundancy and poor global mixing.

**Diagnostic: attention entropy and effective rank.** Given an attention map $\mathbf{A}$ from a layer, we compute its row-wise entropy $H(\mathbf{p}_i)$, where $\mathbf{p}_i$ denotes the $i$-th softmax row. A small entropy indicates that only a few neighboring keys dominate the distribution. At the same time, the Gram matrix $\mathbf{G} = \mathbf{QK}^\top$ tends to be low-rank. To quantify this, we use the *effective rank* defined as $\text{effective\_rank}(\mathbf{M}; \epsilon) = \max\{i \mid \sigma_i(\mathbf{M}) > \epsilon\}$, which discounts negligible singular values and provides a robust proxy for the intrinsic dimensionality. Figure 2 shows that the effective rank remains low on ETTm2, consistent with the qualitative patterns in the heatmaps. We provide further analysis in Appendix F.

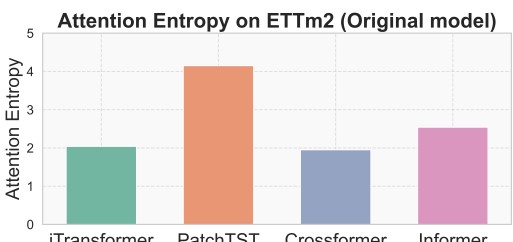

Figure 2: Attention entropy / effective-rank statistics across models on ETTm2. Lower entropy and effective rank signal redundancy and block-like concentration.

### 2.1 CHARACTERIZING BLOCK-WISE ATTENTION

**Why block-wise collapse may occur.** Intuitively, long sequences tend to be highly auto-correlated; keys close to a query index often carry similar information, producing nearly identical logits and attention rows. This effect can be interpreted as a reduction in the effective rank of $\mathbf{QK}^\top$ and a drop in attention entropy. Under such degenerate geometry, optimization updates may become overly aligned across positions, which in turn can slow the model's ability to escape poor local minima.

We now formalize the block-wise attention phenomenon to illustrate its underlying structure. Let $\mathbf{q}_i$ and $\mathbf{k}_j$ be the $i$-th and $j$-th rows of the query and key matrices $\mathbf{Q}$ and $\mathbf{K}$, respectively. Suppose a set of time steps $\mathcal{C} = \{c_1, c_2, \ldots, c_m\}$ exhibit strong temporal correlations such that

$$\text{sim}(\mathbf{q}_{c_i}, \mathbf{k}_{c_j}) \gg \text{sim}(\mathbf{q}_{c_i}, \mathbf{k}_t), \quad \forall i, j, \ \forall t \notin \mathcal{C}, \tag{1}$$

where $\text{sim}(\mathbf{q}, \mathbf{k}) := \mathbf{q}^\top \mathbf{k}$. This condition commonly arises from temporal locality or periodicity in time series data. As a result, attention concentrates within $\mathcal{C}$, producing nearly identical rows in the attention matrix $\mathbf{A}$ and leading to low-rank, block-wise structures. Additionally, inter-variable dependencies can exacerbate this behavior. Multivariate time series often exhibit cross-channel correlations, which compound the locality bias and further reduce the effective rank of $\mathbf{A}$. We next analyze the implications of this structure on optimization.

LOW-RANK STRUCTURE: The rank of the attention matrix $\mathbf{A}$ is upper-bounded by the rank of the query-key product:

$$\mathbf{QK}^\top = \mathbf{XW}_Q \mathbf{W}_K^\top \mathbf{X}^\top, \tag{2}$$

where $\mathbf{Q} = \mathbf{XW}_Q$ and $\mathbf{K} = \mathbf{XW}_K$ with learnable projection matrices $\mathbf{W}_Q, \mathbf{W}_K \in \mathbb{R}^{d \times d_k}$. Thus,

$$\text{rank}(\mathbf{QK}^\top) \leq \min\left(\text{rank}(\mathbf{X}), \ d_k\right). \tag{3}$$

In practice, real-world time series often have significant redundancy and cross-channel dependence (Han et al., 2025), yielding $\text{rank}(\mathbf{X}) \ll L$. Consequently, attention operates over a low-dimensional subspace, resulting in inherently low-rank matrices (Katharopoulos et al., 2020).

IMPACT ON OPTIMIZATION AND GRADIENT FLOW: This low-rank structure influences gradient dynamics during training. Gradients w.r.t. the query and key projection matrices are:

$$\nabla_{\mathbf{W}_Q}\mathcal{L} = \frac{\partial\mathcal{L}}{\partial\mathbf{A}} \cdot \frac{\partial\mathbf{A}}{\partial\mathbf{Q}} \cdot \frac{\partial\mathbf{Q}}{\partial\mathbf{W}_Q}, \quad \nabla_{\mathbf{W}_K}\mathcal{L} = \frac{\partial\mathcal{L}}{\partial\mathbf{A}} \cdot \frac{\partial\mathbf{A}}{\partial\mathbf{K}} \cdot \frac{\partial\mathbf{K}}{\partial\mathbf{W}_K}. \tag{4}$$

If rows of $\mathbf{A}$ are nearly identical due to correlated inputs, their gradients align:

$$\nabla_{\mathbf{W}_Q}\mathcal{L} \approx \sum_{c_i\in\mathcal{C}} \nabla_{\mathbf{A}_{c_i}} \cdot \mathbf{K}, \quad \nabla_{\mathbf{W}_K}\mathcal{L} \approx \sum_{c_i\in\mathcal{C}} \nabla_{\mathbf{A}_{c_i}} \cdot \mathbf{Q}. \tag{5}$$

Such alignment reduces gradient diversity, limiting the model's ability to explore richer representations. As noted by Arora et al. (2018), this can drive convergence to sharp or suboptimal minima, while flat regions (Dinh et al., 2017) prolong training due to weak learning signals. We term this joint degradation of expressivity and optimization *Block-wise Attention Collapse*. This characterization also suggests remedies: reconditioning inputs to break correlations, preserving or raising the effective rank of attention, and encouraging diverse gradients. These principles directly motivate SEAT, introduced next and analyzed with formal guarantees.

**What SEAT introduces.** SEAT is guided by three principles: *(1) Spectral preconditioning.* Applying the DFT projects signals into an *orthogonal* spectral basis, making covariances nearly diagonal and balanced across bands (Gray, 2006; Tilli, 1998). *(2) Near-isometric projection.* A light-weight map on real/imag parts, trained to stay close to orthogonal, acts as a JL/OSE-style embedding, preserving geometry and raising effective rank (Ailon & Chazelle, 2006; Ailon & Liberty, 2009; Sarlós, 2006; Clarkson & Woodruff, 2013). *(3) Group-sparse frequency gating.* Selecting only informative frequency bands reduces variance and aids generalization, echoing structured sparsity (Maurer & Pontil, 2012; Lounici et al., 2011). (see Appendix J for details)

**How SEAT is analyzed theoretically.** Our analysis establishes several formal properties of the proposed approach: *(A) Approximate whitening (Thm. J.1).* Per-frequency scaling by the (estimated) inverse spectral density makes the time-domain covariance close to identity. *(B) Stable-rank preservation (Thm. J.2).* A near-isometric mapping on the whitened subspace preserves, and in some cases increases, the stable rank, helping to mitigate expressivity collapse. *(C) Attention entropy bound (Thm. J.3).* Under isotropic subgaussian assumptions, each attention row satisfies $H(\mathbf{p}_i) \geq \log n - C\sqrt{2\log n}$, ensuring non-trivial entropy. *(D) Gradient diversity (Prop. J.3).* Whitening reduces feature coherence, which in turn lowers expected gradient cosine similarity across positions. *(E) Generalization (Thm. J.4).* Group-sparse spectral selection admits a Rademacher complexity bound of $\tilde{O}(\sqrt{s/n})$, implying improved sample efficiency.

**How SEAT in practice.** Figure 4 illustrates the architecture of the SEAT module used in our experiments. For each channel, we apply FFT, perform per-frequency magnitude normalization, and then pass real and imaginary parts through *dual* linear projections. The transformed features are mapped back to the time domain via inverse FFT, followed by a residual fusion with the input. This module is inserted *before* the attention block (leaving the attention kernel unchanged), adding $O(L\log L)$ computational cost and only negligible parameters. A pseudo-code summary is provided in Algorithm 1 (Appendix B).

## 3 METHODS

**Notation and Input.** Let $X \in \mathbb{R}^{L\times d}$ be an input window (time length $L$, feature/channel dimension $d$). We operate per mini-batch but omit batch indices for clarity. We use a *unitary* real FFT (one-sided): rFFT($X$) returns $K = \lfloor L/2 \rfloor + 1$ frequency bins. We write $\text{Re}(\cdot), \text{Im}(\cdot)$ for real/imag parts and use $\varepsilon = 10^{-6}$ to avoid division by zero.

**Pre-normalization (stationarization).** Before SEAT, we perform window-level stationarization by de-meaning and z-normalizing each feature within the current window. When RevIN is used, we adopt its standard affine scheme and apply the inverse operation (DeNorm) only at the backbone head if required (Lai et al., 2018). This preprocessing makes each window approximately wide-sense stationary (WSS), stabilizing spectral preconditioning.

**Step 1: Domain conversion ("FFT").** We transpose $X$ to $X^\top \in \mathbb{R}^{d\times L}$ and compute a unitary real FFT along time:

$$\hat{X} = \text{rFFT}(X^\top) \in \mathbb{C}^{d\times K}, \quad R = \text{Re}(\hat{X}), \ I = \text{Im}(\hat{X}) \ (R, I \in \mathbb{R}^{d\times K}).$$

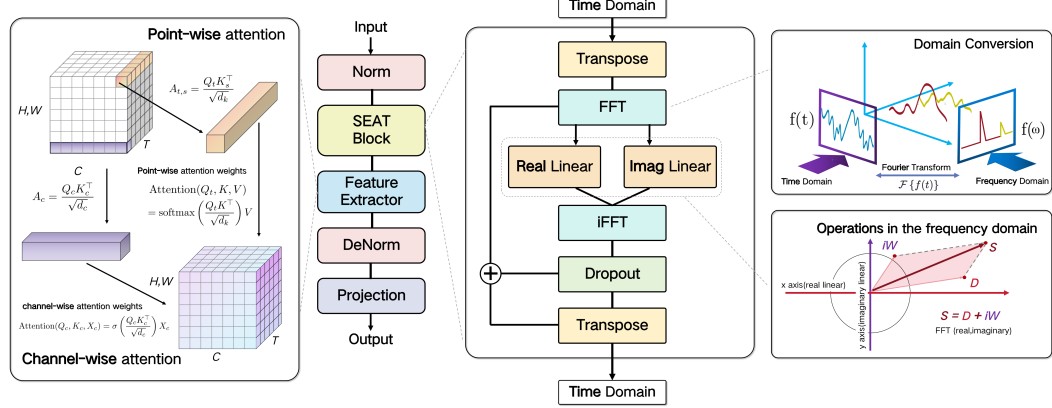

Figure 4: Overview of the SEAT. This projects attention maps into the frequency domain via Fourier transform, applies separate linear transformations on real and imaginary components, and reconstructs the attention using inverse FFT. It acts as a lightweight, plug-and-play module to enhance global temporal dependency modeling in Transformer-based architectures.

**Step 2: Spectral preconditioning (SP).**   We equalize band energy *per frequency across channels*:

$$e_k = \sqrt{\sum_{j=1}^{d}\big(R_{j,k}^2 + I_{j,k}^2\big) + \varepsilon}, \qquad \bar{R}_{j,k} = R_{j,k}/e_k, \ \ \bar{I}_{j,k} = I_{j,k}/e_k.$$

This balances strong/weak harmonics and acts as an approximate whitening step per window (cf. Thm. J.1). (Alternative switch: *per-channel across frequency* normalization, i.e., $e_j = \sqrt{\sum_k(\cdot)}$, gave similar behavior on some datasets.)

**Step 3: Near-isometric dual projection (NIDP; "Real/Imag Linear").**   At each frequency $k$, we linearly mix channels for $\bar{R}_{:,k}$ and $\bar{I}_{:,k}$ with *tied* weights across $k$:

$$Y^R_{:,k} = W_r\,\bar{R}_{:,k} + b_r, \qquad Y^I_{:,k} = W_i\,\bar{I}_{:,k} + b_i,$$

where $W_r, W_i \in \mathbb{R}^{d\times d}$ and $b_r, b_i \in \mathbb{R}^d$. We keep $d \to d$ so that the residual (Step 6) is shape-consistent; if $d \to m(< d)$ is desired, add a time-domain $m \to d$ $1\times 1$ projection before the residual. *Near-isometry.* We encourage $W_\bullet$ to be near-orthogonal via $\mathcal{L}_{\text{ortho}} = \lambda\sum_{\bullet\in\{r,i\}}\|W_\bullet^\top W_\bullet - I\|_F^2$ ($\lambda \in [10^{-4}, 10^{-3}]$) or Householder parameterization. This preserves pairwise geometry and raises the stable rank (Thm. J.2).

**(Optional) Step 4: Spectral Top-$k$ Gate (STG).**   Partition the $K$ positive frequencies into $G$ contiguous groups (default $G = \min\{64, K\}$). Let $g \in \mathbb{R}^G$ be learnable logits (per head or shared). We form a diagonal gate $D = \text{diag}(\pi)$, where $\pi = \text{TopKSoftmax}(g; k = s)$ with temperature $\tau$ (straight-through during training), $s \ll G$. Apply $D$ identically to $Y^R, Y^I$ across channels at the group level. This implements group sparsity and improves small-$n$ generalization (Thm. J.4).

**Step 5: Inverse transform ("iFFT").**   We recompose $\tilde{X}$ in the frequency domain with $Y^R, Y^I$ and perform a unitary inverse transform (one-sided irFFT) to time domain:

$$\widehat{X} = \text{irFFT}\big(Y^R + \mathrm{i}\,Y^I\big)^\top \in \mathbb{R}^{L\times d}.$$

**Step 6: Residual fusion & dropout.**   We add the input residual with a learnable scalar (or per-channel) scale $\alpha$ and optional dropout $p$, and $Z$ is fed to the attention block (we do not modify the attention kernel):

$$Z = X + \alpha \cdot \widehat{X}, \quad \alpha \in \mathbb{R} \ (\text{default } 1.0), \ \ p \in [0, 0.2].$$

**Learnable parameters (per SEAT block).**

$$\Theta_{\text{SEAT}} = \{W_r, W_i, b_r, b_i, \ g \text{ (if STG)}, \ \alpha\}.$$

By default we *share $W_r, W_i$ over all frequencies $k$* and *do not share across layers*. Biases $b_\bullet$ are optional (0-init). Initialization: $W_\bullet$ from identity + orthogonal noise ($\sigma \approx 10^{-3}$) or orthogonal init; $g$ zero-init; $\alpha = 1$.

## 4 EXPERIMENTS

We empirically evaluate SEAT on standard time series forecasting benchmarks. Subsection 4.1 reports the main results comparing SEAT with state-of-the-art Transformer architectures. Subsection 4.2 evaluates SEAT's plug-and-play compatibility with diverse backbones. Subsection 4.3 presents ablation studies analyzing SEAT's effect on optimization dynamics, while Subsection 4.4 visualizes attention patterns to highlight SEAT's representational benefits. Finally, we provide a detailed analysis of SEAT's time complexity in Appendix E.

**Datasets:** We evaluate on eight widely used multivariate time series forecasting datasets: ETTh1, ETTh2, ETTm1, and ETTm2 (Zhou et al., 2021); Weather, Exchange, ECL (Electricity), and Traffic (Lai et al., 2018). We follow standard preprocessing and evaluation protocols established in prior work (Zhou et al., 2021; Zeng et al., 2023; Hebrail & Berard, 2012). Detailed dataset statistics and preprocessing procedures are provided in Appendix A.

**Baselines:** To evaluate generality, we integrate SEAT into seven leading Transformer architectures: iTransformer (Liu et al., 2024a), PatchTST (Nie et al., 2023), Crossformer (Zhang & Yan, 2023), Pyraformer (Liu et al., 2022a), Autoformer (Wu et al., 2021), Informer (Zhou et al., 2021), and Reformer (Kitaev et al., 2020). All models are trained with Reversible Instance Normalization (RevIN) (Kim et al., 2021) for fairness. Unless otherwise specified, we use iTransformer as the default backbone when reporting SEAT's results. Further implementation and training details are given in Appendix A.

### 4.1 MAIN RESULTS

Table 1 summarizes forecasting results, with the best and second-best scores marked in **red** and blue, respectively. Lower MSE and MAE indicate better accuracy. When equipped with iTransformer as the backbone, SEAT achieves state-of-the-art performance, ranking first on all 16 metrics across eight benchmark datasets. The gains are especially pronounced on challenging datasets such as ECL and Exchange. For instance, on the ECL dataset, SEAT reduces MSE to 0.167 and MAE to 0.257, yielding relative improvements of 6.2% in MSE and 4.8% in MAE compared to the strongest baseline (iTransformer). Similarly, on the Exchange dataset, SEAT improves MSE by 10.0% and MAE by 3.2% over iTransformer. These results highlight that augmenting iTransformer with SEAT not only improves overall forecasting accuracy but also provides robustness on highly variable and non-stationary datasets where conventional attention mechanisms typically struggle.

These results validate SEAT's spectral decomposition and frequency-aware attention, which enhance global temporal modeling and suppress local noise, improving representation and accuracy. Crucially, SEAT achieves this without altering backbone architectures, confirming its lightweight, plug-and-play design. Its consistent top performance across benchmarks underscores the importance of spectral inductive biases in advancing attention-based temporal forecasting. SEAT enables Transformer models to handle long-range dependencies and high-dimensional inputs more effectively while mitigating block-wise attention. Detailed results and additional settings are provided in Appendix C.1.

### 4.2 PLUG-AND-PLAY INTEGRATION EXPERIMENTS

To evaluate SEAT's generalization ability, we integrated it into a range of Transformer architectures and measured forecasting accuracy across standard benchmarks. As shown in Table 2, SEAT consistently improves performance, with notable MSE reductions such as 43.6% for Pyraformer and 38.8% for Reformer on ETTh1, and gains exceeding 80% for Informer and Reformer on ETTh2. Two complementary mechanisms underlie these gains: a spectral conditioning step that balances energy across frequencies and steadies temporal statistics, and—crucially—an enhancement of the attention

Table 1: Mean performance comparison between SEAT and other state-of-the-art Transformer models. On eight benchmark datasets, SEAT achieves MSE and MAE in all cases, demonstrating consistent superiority in forecasting accuracy.

| Models | SEAT Ours | | iTransformer ICLR 2024 | | PatchTST ICLR 2023 | | Crossformer ICLR 2023 | | Pyraformer ICLR 2023 | | Autoformer Neurips 2022 | | Informer AAAI 2021 | | Reformer ICLR 2020 | |
|---|---|---|---|---|---|---|---|---|---|---|---|---|---|---|---|---|
| Metric | MSE | MAE | MSE | MAE | MSE | MAE | MSE | MAE | MSE | MAE | MSE | MAE | MSE | MAE | MSE | MAE |
| ETTh1 | **0.444** | **0.431** | 0.454 | 0.447 | 0.469 | 0.454 | 0.529 | 0.522 | 0.865 | 0.731 | 0.518 | 0.500 | 1.078 | 0.813 | 0.961 | 0.757 |
| ETTh2 | **0.367** | **0.395** | 0.383 | 0.407 | 0.387 | 0.407 | 0.942 | 0.684 | 3.755 | 1.551 | 0.432 | 0.451 | 3.490 | 1.532 | 3.574 | 1.525 |
| ETTm1 | **0.380** | **0.384** | 0.407 | 0.410 | 0.387 | 0.400 | 0.513 | 0.496 | 0.750 | 0.615 | 0.583 | 0.513 | 0.948 | 0.717 | 0.928 | 0.688 |
| ETTm2 | **0.277** | **0.318** | 0.288 | 0.332 | 0.281 | 0.326 | 0.757 | 0.610 | 1.509 | 0.845 | 0.332 | 0.370 | 1.489 | 0.867 | 1.415 | 0.862 |
| Weather | **0.244** | **0.265** | 0.258 | 0.278 | 0.259 | 0.281 | 0.259 | 0.315 | 0.278 | 0.342 | 0.317 | 0.359 | 0.723 | 0.605 | 0.485 | 0.500 |
| ECL | **0.167** | **0.257** | 0.178 | 0.270 | 0.205 | 0.290 | 0.244 | 0.334 | 0.298 | 0.389 | 0.230 | 0.339 | 0.377 | 0.449 | 0.302 | 0.392 |
| Exchange | **0.324** | **0.390** | 0.360 | 0.403 | 0.367 | 0.404 | 0.940 | 0.707 | 1.308 | 0.945 | 0.493 | 0.493 | 1.411 | 0.968 | 1.000 | 0.837 |
| Traffic | **0.423** | **0.266** | 0.428 | 0.282 | 0.481 | 0.304 | 0.550 | 0.304 | 1.185 | 0.553 | 0.761 | 0.479 | 0.868 | 0.472 | 0.648 | 0.347 |
| 1st count | **8** | **8** | 0 | 0 | 0 | 0 | 0 | 0 | 0 | 0 | 0 | 0 | 0 | 0 | 0 | 0 |

Table 2: Performance comparison of Transformer-based forecasting models before and after integrating the SEAT plugin on eight datasets. Metrics include MSE and MAE. Relative improvements brought by SEAT are highlighted in **red**.

| | Models Ours | iTransformer ICLR 2024 | | PatchTST ICLR 2023 | | Crossformer ICLR 2023 | | Pyraformer ICLR 2023 | | Autoformer NeurIPS 2022 | | Informer AAAI 2021 | | Reformer ICLR 2020 | |
|---|---|---|---|---|---|---|---|---|---|---|---|---|---|---|---|
| | Metric | MSE | MAE | MSE | MAE | MSE | MAE | MSE | MAE | MSE | MAE | MSE | MAE | MSE | MAE |
| ETTh1 | Original | 0.454 | 0.447 | 0.469 | 0.454 | 0.529 | 0.522 | 0.865 | 0.731 | 0.518 | 0.500 | 1.078 | 0.813 | 0.961 | 0.757 |
| | +SEAT | 0.443 | 0.431 | 0.447 | 0.443 | 0.484 | 0.466 | 0.488 | 0.474 | 0.486 | 0.468 | 1.026 | 0.736 | 0.588 | 0.530 |
| | Improvement | +2.4% | +3.6% | +4.7% | 2.6% | +8.5% | +10.7% | +43.6% | +35.2% | +6.2% | +6.4% | +4.8% | +9.5% | +38.8% | +29.9% |
| ETTh2 | Original | 0.383 | 0.407 | 0.387 | 0.407 | 0.942 | 0.684 | 3.755 | 1.551 | 0.432 | 0.451 | 3.490 | 1.532 | 3.574 | 1.525 |
| | +SEAT | 0.367 | 0.395 | 0.374 | 0.402 | 0.394 | 0.416 | 0.433 | 0.434 | 0.459 | 0.449 | 0.677 | 0.571 | 0.459 | 0.451 |
| | Improvement | +4.2% | +2.9% | +3.4% | 1.2% | +58.2% | +39.1% | +88.5% | +72.0% | -6.3% | +0.4% | +80.6% | +62.7% | +87.2% | +70.4% |
| ETTm1 | Original | 0.407 | 0.410 | 0.387 | 0.400 | 0.513 | 0.496 | 0.750 | 0.615 | 0.583 | 0.513 | 0.948 | 0.717 | 0.928 | 0.688 |
| | +SEAT | 0.380 | 0.384 | 0.380 | 0.396 | 0.410 | 0.411 | 0.427 | 0.425 | 0.529 | 0.480 | 0.629 | 0.530 | 0.599 | 0.507 |
| | Improvement | +6.6% | +6.3% | +1.8% | 1.0% | +20.1% | +17.0% | +43.0% | +30.9% | +9.3% | +6.4% | +33.6% | +26.1% | +35.4% | +26.3% |
| ETTm2 | Original | 0.288 | 0.332 | 0.281 | 0.326 | 0.757 | 0.610 | 1.509 | 0.845 | 0.332 | 0.370 | 1.489 | 0.867 | 1.415 | 0.862 |
| | +SEAT | 0.277 | 0.318 | 0.279 | 0.326 | 0.292 | 0.332 | 0.302 | 0.336 | 0.304 | 0.342 | 0.389 | 0.406 | 0.318 | 0.348 |
| | Improvement | +3.8% | +4.2% | +0.7% | +0.2% | +61.4% | +45.7% | +80.0% | +60.2% | +8.4% | +7.6% | +73.9% | +53.2% | +77.5% | +59.6% |
| weather | Original | 0.258 | 0.278 | 0.259 | 0.281 | 0.259 | 0.315 | 0.278 | 0.342 | 0.317 | 0.359 | 0.723 | 0.605 | 0.485 | 0.500 |
| | +SEAT | 0.244 | 0.265 | 0.254 | 0.281 | 0.261 | 0.292 | 0.257 | 0.284 | 0.278 | 0.304 | 0.289 | 0.320 | 0.273 | 0.299 |
| | Improvement | +5.4% | +4.7% | +1.9% | -0.4% | -0.8% | +7.3% | +7.6% | +17.0% | +12.3% | +15.3% | +60.0% | +47.1% | +43.7% | +40.2% |
| ECL | Original | 0.178 | 0.270 | 0.205 | 0.290 | 0.244 | 0.334 | 0.298 | 0.389 | 0.230 | 0.339 | 0.377 | 0.449 | 0.302 | 0.392 |
| | +SEAT | 0.167 | 0.257 | 0.186 | 0.277 | 0.167 | 0.260 | 0.206 | 0.309 | 0.208 | 0.308 | 0.258 | 0.358 | 0.210 | 0.312 |
| | Improvement | +6.2% | +4.8% | +9.3% | 4.5% | +31.6% | +22.2% | +30.9% | +20.6% | +9.6% | +9.1% | +31.6% | +20.3% | +30.2% | +20.4% |
| Exchange | Original | 0.360 | 0.403 | 0.367 | 0.404 | 0.940 | 0.707 | 1.308 | 0.945 | 0.493 | 0.493 | 1.411 | 0.968 | 1.000 | 0.837 |
| | +SEAT | 0.324 | 0.390 | 0.365 | 0.405 | 0.367 | 0.414 | 0.396 | 0.429 | 0.459 | 0.466 | 0.368 | 0.428 | 0.448 | 0.459 |
| | Improvement | +10.0% | +3.2% | +0.3% | -0.2% | +61.0% | +41.4% | +69.7% | +54.6% | +6.9% | +5.5% | +73.9% | +55.8% | +55.2% | +45.2% |
| traffic | Original | 0.428 | 0.282 | 0.481 | 0.304 | 0.550 | 0.304 | 1.185 | 0.553 | 0.761 | 0.479 | 0.868 | 0.472 | 0.648 | 0.347 |
| | +SEAT | 0.423 | 0.266 | 0.477 | 0.291 | 0.479 | 0.311 | 0.794 | 0.436 | 0.713 | 0.392 | 1.030 | 0.567 | 0.638 | 0.333 |
| | Improvement | +1.2% | +5.7% | +0.8% | +4.3% | +12.9% | -2.3% | +33.0% | +21.16% | +6.2% | +18.2% | -18.7% | -20.1% | +1.4% | +4.0% |

matrix's effective rank that counters block-wise collapse and expands the usable attention subspace. The latter proves especially consequential for accuracy: it preserves discriminative structure across long horizons and variables even when temporal statistics are already stabilized by stationarization modules (e.g., RevIN or non-stationary modeling). On datasets with pronounced seasonality, such as ECL and Exchange, this combination is particularly effective, with Crossformer and Pyraformer improving by over 30% and 60% MSE, respectively. On Traffic, Pyraformer and Autoformer benefit substantially (+33.0% and +6.2% MSE), whereas Informer shows mild degradation, indicating possible architectural incompatibilities. Notably, architectures that already include normalization or stationarization (e.g., iTransformer and PatchTST) still improve with SEAT, suggesting that temporal stabilization alone does not remove residual low-rank bias, whereas rank enhancement directly addresses this bottleneck. Detailed, dataset-specific results are presented in Appendix C.2, offering a comprehensive overview of SEAT's performance across various forecasting scenarios.

## 4.3 ABLATION STUDIES

To rigorously assess each component's contribution in SEAT, we conduct ablation studies on two variants: (1) SEAT w/o FFT, removing frequency-domain decomposition and applying rank enhancement

in the time domain; and (2) SEAT w/o Residual, retaining frequency-based filtering but omitting the residual connection.

As Table 3 shows, both variants consistently underperform the full model. The significant drop in SEAT w/o FFT confirms that time-domain operations alone cannot effectively disentangle long-range dependencies and noise, highlighting the necessity of spectral decomposition. The milder decline in SEAT w/o Residual underscores the residual connection's role in stabilizing optimization, preventing overfitting and loss of global context. These findings demonstrate that frequency-domain transformation and residual connections are complementary and indispensable for SEAT's strong performance, balancing selective feature suppression with temporal trend preservation.

Table 3: Ablation study on SEAT. "w/o FFT" removes frequency-domain decomposition. "w/o Residual" removes the residual connection.

| Method | ETTh1 | | ETTh2 | | ECL | | Exchange | | Traffic | |
|--------|-------|-------|-------|-------|-------|-------|-------|-------|-------|-------|
| Metrics | MSE | MAE | MSE | MAE | MSE | MAE | MSE | MAE | MSE | MAE |
| **SEAT w/o FFT** | 0.473 | 0.461 | 0.417 | 0.429 | 0.229 | 0.317 | 0.390 | 0.421 | 0.458 | 0.310 |
| **SEAT w/o Residual** | 0.455 | 0.447 | 0.397 | 0.417 | 0.196 | 0.283 | 0.368 | 0.409 | 0.429 | 0.288 |
| **SEAT** | **0.444** | **0.431** | **0.367** | **0.395** | **0.167** | **0.256** | **0.324** | **0.390** | **0.423** | **0.266** |

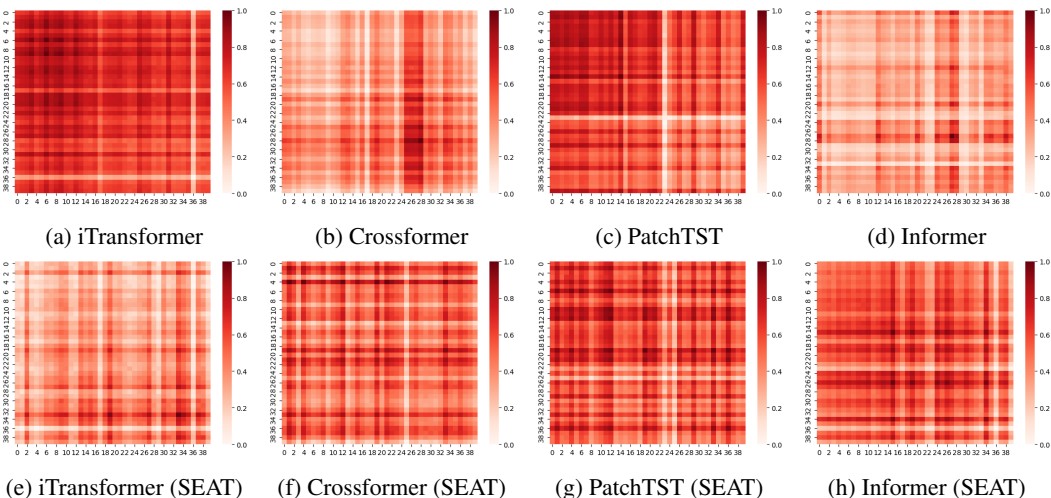

(a) iTransformer     (b) Crossformer     (c) PatchTST     (d) Informer

(e) iTransformer (SEAT)     (f) Crossformer (SEAT)     (g) PatchTST (SEAT)     (h) Informer (SEAT)

Figure 5: Attention heatmaps of the original model (a–d) and the model enhanced with SEAT (e–h) on the ETTm2 dataset. The original model exhibits obvious block-wise attention patterns with low discriminability. After introducing SEAT, the block structure is alleviated and the attention maps show improved focus and sharper distinctions, indicating better global awareness and feature separation.

## 4.4 MODEL IMPACT VISUALIZATION

To assess SEAT's effectiveness, we visualize attention heatmaps before and after its integration (Figure 5). In the original models, attention exhibits block-wise structures with strong diagonal dominance, reflecting over-reliance on adjacent tokens and weak token discrimination. This collapse limits representational flexibility and long-range modeling. With SEAT, attention becomes sharper and more globally distributed, consistent with its design principle of spectral preconditioning: by whitening input statistics in the frequency domain, SEAT balances energy across modes, suppresses dominant directions, and raises the stable rank of the attention map. As a result, allocations spread more evenly across tokens, improving discrimination and diversity.

To quantify these changes, we measure the effective rank and energy ratio of the top five singular values of attention matrices (Figure 6). Across all tested models, SEAT achieves higher effective ranks, showing that attention weights occupy a broader set of singular modes instead of collapsing into a low-dimensional subspace. Meanwhile, lower energy concentration in the top-5 singular values indicates a more balanced spectrum, aligning with the claim that whitening alleviates low-rank bias

and enriches representational capacity. These results confirm that spectral conditioning enlarges the usable attention subspace and mitigates rank collapse.

We also track the diagonal weight ratio of attention matrices (Figure 7). High diagonal ratios correspond to short-range, block-wise attention that hinders long-range dependency modeling. The original models—especially iTransformer—show increasing diagonal dominance during training, signaling drift toward local focus. In contrast, SEAT substantially reduces and stabilizes diagonal ratios, particularly in Crossformer and iTransformer, thereby maintaining balanced and global attention distributions throughout optimization.

Together, the visualizations and metrics demonstrate that SEAT suppresses block-wise collapse, increases attention diversity, and strengthens long-range dependency modeling. By linking spectral whitening to higher effective rank and reduced low-rank bias, these empirical results directly support our theoretical analysis. Extended results and visualizations appear in Appendix F.

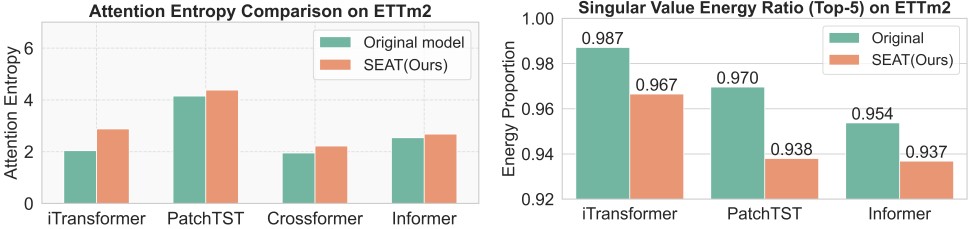

Figure 6: (a) Attention entropy between original models and their SEAT-augmented counterparts on ETTm2 dataset (Left). Lower entropy indicates less diverse attention, suggesting reduced representational capacity. SEAT enhances representational diversity by increasing attention entropy.(b) Comparison of the energy ratio captured by the top-5 singular values of attention matrices across different models on the ETTm2 dataset (Right). Models equipped with the SEAT (Ours) module exhibit lower energy concentration in the top singular values, indicating reduced low-rank bias and improved representational richness in the attention mechanism.

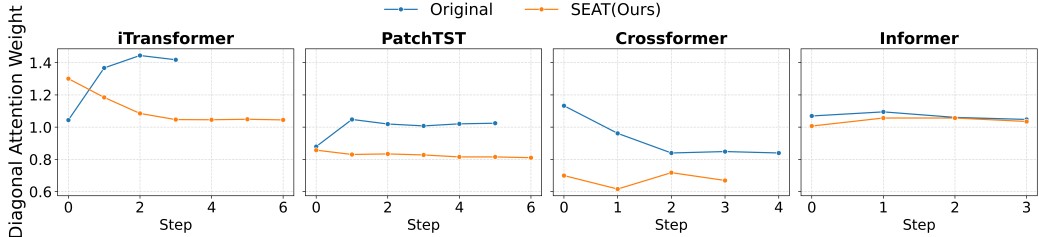

Figure 7: Diagonal attention weight ratios over training steps across Transformer-based models. Higher ratios indicate stronger local attention. Baseline models (blue) show increasing diagonal dominance, reflecting local overfitting. SEAT-augmented models (orange) maintain lower, stable ratios, indicating reduced locality bias and improved modeling of long-range dependencies.

## 5 CONCLUSION

We introduce **SEAT**, a spectral-enhanced attention module designed to alleviate block-wise collapse in Transformer-based time series forecasting. Our theoretical analysis establishes that standard attention can degenerate into low-rank patterns under strong temporal and cross-variable correlations, which limits expressiveness and hinders optimization. By contrast, the proposed module provides formal guarantees on whitening, stable-rank preservation, and gradient diversity, ensuring more robust representational capacity. Being lightweight and model-agnostic, it can be incorporated into existing Transformer architectures with minimal overhead. Empirically, SEAT achieves consistent improvements in forecasting accuracy across eight benchmark datasets. Moreover, attention entropy analysis confirms that the method induces richer and more diverse attention patterns, thereby validating our theoretical insights and offering guidance for future attention design.

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

## A    Detailed Experimental Settings

### A.1    Datasets

**Datasets.** We conducted extensive experiments on eight widely-used benchmark datasets to evaluate the performance of baseline models and the proposed SEAT framework. The ETT dataset (Zhou et al., 2021) contains seven features related to electricity transformer temperatures, sampled at different frequencies. It is divided into four subsets: ETTh1 and ETTh2 are recorded hourly, while ETTm1 and ETTm2 are recorded every 15 minutes, covering the period from July 2016 to July 2018. The Weather dataset (Lai et al., 2018) includes 21 meteorological variables collected every 10 minutes in 2020 from the Max Planck Institute for Biogeochemistry. The Exchange dataset (Lai et al., 2018) consists of daily exchange rates from eight countries spanning from 1990 to 2016. The ECL (Electricity) dataset (Lai et al., 2018) records hourly electricity consumption from 321 clients. Lastly, the Traffic dataset (Lai et al., 2018) captures hourly road occupancy rates from 862 sensors located on freeways in the San Francisco Bay Area, covering January 2015 to December 2016. These datasets span diverse domains such as energy systems, finance, meteorology, and transportation, posing a variety of forecasting challenges—particularly in long-horizon settings. Detailed statistics are summarized in Table 4.

**Dataset Split.** We follow the data processing procedures and chronological data split strategy adopted in TimesNet (Wu et al., 2023). For all datasets, the training, validation, and test sets are partitioned sequentially without shuffling to preserve temporal dependencies. Specifically, we apply a (0.7, 0.1, 0.2) split ratio for Weather, Traffic, ECL, and Exchange datasets, and a (0.6, 0.2, 0.2) split ratio for the ETT subsets (ETTh1, ETTh2, ETTm1, and ETTm2), consistent with prior literature (Zhou et al., 2021; Zeng et al., 2023). This setup ensures a fair and standardized evaluation protocol, especially for long-term forecasting benchmarks.

**Forecasting setting.** We standardize the experimental configurations across all models to ensure a fair comparison within a unified framework. Specifically, the input sequence length is fixed to 96, while the prediction horizons vary among {96, 192, 336, 720}.

Table 4: The summary of dataset.

| Dataset | Dim | Time Points | Frequency | Information |
|---------|-----|-------------|-----------|-------------|
| ETTh1   | 7   | 17,420      | 1-hour    | Electricity |
| ETTh2   | 7   | 17,420      | 1-hour    | Electricity |
| ETTm1   | 7   | 69,680      | 15-min    | Electricity |
| ETTm2   | 7   | 69,680      | 15-min    | Electricity |
| ECL     | 321 | 26,304      | 1-hour    | Electricity |
| Traffic | 862 | 17,544      | 1-hour    | Transportation |
| Weather | 21  | 52,696      | 10-min    | Weather |
| Solar   | 137 | 52,560      | 10-min    | Energy |
| Exchange| 8   | 7,588       | 1-day     | Financial |

### A.2    Experiment Setting

We standardized the parameters across all models to ensure a fair comparison within a unified platform (Time-Series-Library). Specifically, in main experiment, the batch size (`batch_size`) was set to 16. To prevent overfitting, a dropout rate of 0.1 was applied. The number of training epochs was fixed at 30. For temporal encoding, we adopted the `timeF` embedding method. Additional parameter details are provided in Table 5. Experiments are implemented in PyTorch (Paszke et al., 2019) and executed on a single NVIDIA RTX 4090 GPU (24GB). In the plug-and-play experiment, we adopt the experimental settings as defined in the original papers of the seven baseline models.

Table 5: Experiment configuration of SEAT.

| Dataset | ETTh1 | ETTh2 | ETTm1 | ETTm2 | Weather | Electricity | Exchange | Traffic |
|---------|-------|-------|-------|-------|---------|-------------|----------|---------|
| seq_len | 96 | 96 | 96 | 96 | 96 | 96 | 96 | 96 |
| label_len | 48 | 48 | 48 | 48 | 96 | 48 | 96 | 48 |
| e_layers | 2 | 2 | 2 | 2 | 2 | 2 | 2 | 2 |
| d_layers | 1 | 1 | 1 | 1 | 1 | 1 | 1 | 1 |
| factor | 3 | 3 | 1 | 1 | 3 | 3 | 3 | 3 |
| enc_in | 7 | 7 | 7 | 7 | 21 | 321 | 8 | 862 |
| dec_in | 7 | 7 | 7 | 7 | 21 | 321 | 8 | 862 |
| c_out | 7 | 7 | 7 | 7 | 21 | 321 | 8 | 862 |
| d_model | 512 | 512 | 512 | 512 | 32 | 512 | 64 | 512 |
| d_ff | – | 16 | – | – | 32 | – | 64 | – |
| n_heads | 8 | 8 | 8 | 8 | 8 | 8 | 8 | 8 |
| lr | 1e-4 | 1e-4 | 1e-4 | 1e-4 | 1e-4 | 1e-4 | 1e-4 | 1e-3 |

## B  ALGORITHM OF SEAT

---

**Algorithm 1** SEAT: Sparsity-Enhanced Attention Transformer Module

---

1: **Input:** Input sequence $\mathbf{X} \in \mathbb{R}^{T \times d}$
2: **Output:** Enhanced sequence $\mathbf{Z} \in \mathbb{R}^{T \times d}$
3: */* Apply FFT independently on each feature dimension */
4: **for** $j = 1, \dots, d$ **do**
5:    **for** $k = 0, \dots, T-1$ **do**
6:       $\mathbf{X}_f^{(j)}[k] \leftarrow \sum_{t=0}^{T-1} X[t,j] \cdot e^{-j 2\pi kt/T}$
7:    **end**
8: **end**
9: */* Normalize each frequency vector */
10: **for** $j = 1, \dots, d$ **do**
11:    $\mathcal{A}_f^{(j)} \leftarrow \left( \left| \mathcal{X}_f^{(j)}[0] \right|, \dots, \left| \mathcal{X}_f^{(j)}[T-1] \right| \right)$
12:    **for** $k = 0, \dots, T-1$ **do**
13:       $\tilde{\mathcal{X}}_f^{(j)}[k] \leftarrow \mathcal{X}_f^{(j)}[k] / \|\mathcal{A}_f^{(j)}\|_2$
14:    **end**
15: **end**
16: */* Separate real and imaginary parts */
17: $\text{Re}(\tilde{\mathcal{X}}_f), \text{Im}(\tilde{\mathcal{X}}_f) \in \mathbb{R}^{T \times d}$
18: */* Apply learned linear projections */
   $\widehat{\text{Re}}(\mathcal{X}_f) \leftarrow \mathbf{W}_r \cdot \text{Re}(\tilde{\mathcal{X}}_f) + b_r$
   $\widehat{\text{Im}}(\mathcal{X}_f) \leftarrow \mathbf{W}_i \cdot \text{Im}(\tilde{\mathcal{X}}_f) + b_i$
19: */* Form enhanced complex signal */
20: $\widehat{\mathcal{X}}_f[k] \leftarrow \widehat{\text{Re}}(\mathcal{X}_f)[k] + j \cdot \widehat{\text{Im}}(\mathcal{X}_f)[k]$
21: */* Transform back to time domain */
22: **for** $t = 0, \dots, T-1$ **do**
23:    $\widehat{\mathbf{X}}[t] \leftarrow \frac{1}{T} \sum_{k=0}^{T-1} \widehat{\mathcal{X}}_f[k] \cdot e^{j 2\pi kt/T}$
24: **end**
25: */* Residual fusion */
26: $\mathbf{Z} \leftarrow \mathbf{X} + \widehat{\mathbf{X}}$
27: **return Z**

---

## C  FULL RESULTS

### C.1  MAIN RESULTS

We evaluate the forecasting performance of our proposed method across eight widely used multivariate time series datasets: ETTh1, ETTh2, ETTm1, ETTm2, Electricity, Exchange, Traffic, and Weather. The evaluation is conducted under four different prediction horizons (96, 192, 336, and 720), using Mean Squared Error (MSE) and Mean Absolute Error (MAE) as performance metrics. To assess the effectiveness of the proposed SEAT module, we adopt iTransformer as the default backbone

and integrate SEAT into it. The resulting model is then compared with a range of state-of-the-art Transformer-based forecasting models.

Table 6: Forecasting performance (MSE and MAE) on eight datasets under different prediction horizons.

| Dataset | ETTh1 | | ETTh2 | | ETTm1 | | ETTm2 | | Electricity | | Exchange | | Traffic | | Weather | |
|---|---|---|---|---|---|---|---|---|---|---|---|---|---|---|---|---|
| metric | MSE | MAE | MSE | MAE | MSE | MAE | MSE | MAE | MSE | MAE | MSE | MAE | MSE | MAE | MSE | MAE |
| 96 | 0.383 | 0.393 | 0.286 | 0.338 | 0.310 | 0.341 | 0.174 | 0.252 | 0.139 | 0.228 | 0.084 | 0.204 | 0.390 | 0.251 | 0.158 | 0.195 |
| 192 | 0.434 | 0.423 | 0.360 | 0.386 | 0.363 | 0.370 | 0.239 | 0.295 | 0.157 | 0.245 | 0.170 | 0.297 | 0.416 | 0.259 | 0.208 | 0.241 |
| 336 | 0.471 | 0.441 | 0.401 | 0.420 | 0.395 | 0.392 | 0.299 | 0.334 | 0.170 | 0.259 | 0.299 | 0.403 | 0.425 | 0.269 | 0.267 | 0.286 |
| 720 | 0.487 | 0.466 | 0.420 | 0.438 | 0.453 | 0.431 | 0.395 | 0.390 | 0.202 | 0.289 | 0.744 | 0.656 | 0.460 | 0.286 | 0.345 | 0.337 |
| Avg. | 0.444 | 0.431 | 0.367 | 0.395 | 0.380 | 0.384 | 0.277 | 0.318 | 0.167 | 0.256 | 0.324 | 0.390 | 0.423 | 0.266 | 0.244 | 0.265 |

## C.2 PLUG-AND-PLAY

We apply the SEAT framework to Transformer variants such as Reformer (Kitaev et al., 2020), Informer (Zhou et al., 2021), and iTransformer(Liu et al., 2024a). The forecasting results, as shown in Table 7, demonstrate that our SEAT framework consistently enhances the performance of these Transformer variants across various benchmarks.

Table 7: Performance comparison in terms of forecasting error metrics. This table illustrates the performance comparison among different models in terms of forecasting error metrics, adhering to a unified setting to ensure fairness. The mse and mae values highlight the accuracy of the predictions. The best-performing results are highlighted in **red**, while the second-best results are marked in blue with underlining. Lower MSE/MAE values signify higher predictive accuracy. The incorporation of SEAT into various benchmark attention models demonstrates significant performance enhancements, showcasing SEAT's effectiveness in improving the forecasting capabilities of Transformer-based models as a model-agnostic plugin.

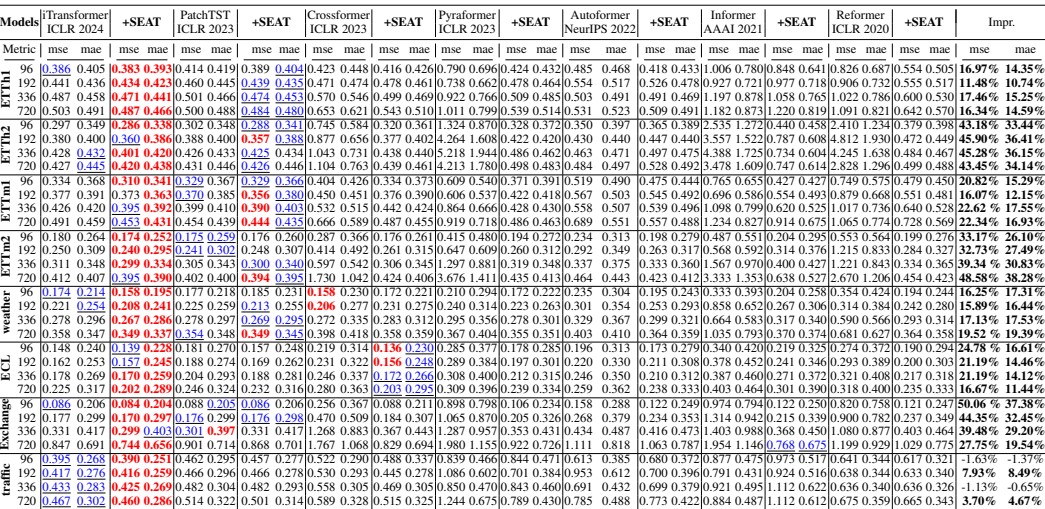

## C.3 ABLATION STUDY

To validate the effectiveness of the SEAT module, we conduct a comprehensive ablation study. Specifically, we examine two reduced variants: (1) SEAT w/o FFT, which removes the frequency-domain decomposition and performs rank enhancement directly in the time domain; and (2) SEAT w/o Residual, which retains frequency-domain operations but excludes the residual connection.

As shown in Table 8, both ablated versions consistently underperform the full SEAT model across all datasets. This confirms that both the frequency-domain rank enhancement and the residual pathway are essential components contributing to the final performance. In particular, removing the FFT module leads to more significant degradation, highlighting the benefit of decorrelating temporal patterns via spectral decomposition.

Table 8: Ablation study on SEAT without FFT and without Residual connection across five datasets. Metrics: MSE and MAE.

| Dataset | Len | SEAT_w/o_FFT | | SEAT_w/o_Residual | | SEAT | |
|---|---|---|---|---|---|---|---|
| | | MSE | MAE | MSE | MAE | MSE | MAE |
| ETTh1 | 96 | 0.406 | 0.422 | 0.389 | 0.408 | 0.383 | 0.393 |
| | 192 | 0.465 | 0.455 | 0.438 | 0.435 | 0.434 | 0.423 |
| | 336 | 0.493 | 0.466 | 0.483 | 0.455 | 0.471 | 0.441 |
| | 720 | 0.530 | 0.501 | 0.510 | 0.491 | 0.487 | 0.466 |
| ETTh2 | 96 | 0.314 | 0.361 | 0.341 | 0.378 | 0.286 | 0.338 |
| | 192 | 0.401 | 0.413 | 0.414 | 0.421 | 0.360 | 0.386 |
| | 336 | 0.437 | 0.442 | 0.465 | 0.457 | 0.401 | 0.420 |
| | 720 | 0.435 | 0.451 | 0.448 | 0.460 | 0.420 | 0.438 |
| ECL | 96 | 0.169 | 0.259 | 0.204 | 0.296 | 0.139 | 0.228 |
| | 192 | 0.180 | 0.269 | 0.209 | 0.301 | 0.157 | 0.245 |
| | 336 | 0.197 | 0.286 | 0.230 | 0.320 | 0.170 | 0.259 |
| | 720 | 0.237 | 0.319 | 0.272 | 0.352 | 0.202 | 0.289 |
| Exchange | 96 | 0.090 | 0.212 | 0.091 | 0.214 | 0.084 | 0.204 |
| | 192 | 0.181 | 0.304 | 0.185 | 0.307 | 0.170 | 0.297 |
| | 336 | 0.330 | 0.417 | 0.339 | 0.422 | 0.299 | 0.403 |
| | 720 | 0.871 | 0.705 | 0.945 | 0.741 | 0.744 | 0.656 |
| Traffic | 96 | 0.400 | 0.274 | 0.428 | 0.299 | 0.390 | 0.251 |
| | 192 | 0.421 | 0.283 | 0.444 | 0.302 | 0.416 | 0.259 |
| | 336 | 0.427 | 0.287 | 0.462 | 0.310 | 0.425 | 0.269 |
| | 720 | 0.469 | 0.307 | 0.496 | 0.329 | 0.460 | 0.286 |

## D    HYPERPAPAMETER SENSITIVITY

To evaluate the robustness and adaptability of the proposed SEAT framework under different hyperparameter configurations, we conduct a comprehensive sensitivity analysis in Figure 8. Specifically, we vary four key hyperparameters—learning rate (lr), batch size, number of encoder layers (e_layers), and model dimension (d_model)—across four widely-used multivariate time series datasets: ETTh1, ETTm1, ECL, and Exchange. The Mean Squared Error (MSE) is adopted as the evaluation metric.

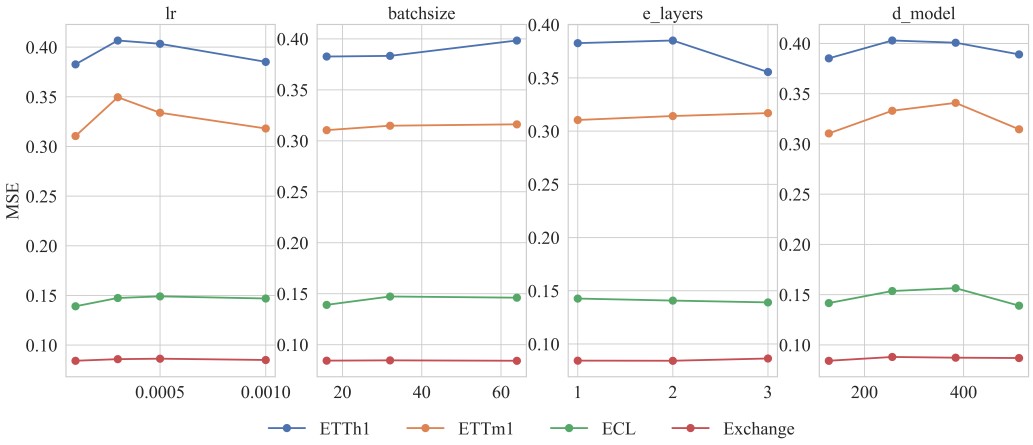

Figure 8: Hyperparameter sensitivity analysis of the SEAT framework on four benchmark datasets (ETTh1, ETTm1, ECL, Exchange). We vary the learning rate, batch size, number of encoder layers, and model dimension, and report the corresponding MSE. Results demonstrate the robustness and stable performance of SEAT across different configurations.

## E    TIME COMPLEXITY

As SEAT is designed as a plugin to enhance the transformer model, its additional computational complexity is minimal. Instead, it leverages frequency-domain processing (FFT) and sparsity techniques that reduce the overall complexity, demonstrating that the additional computational cost of incorporating SEAT is relatively low.

**Complexity (per mini-batch of size $B$).**   Let $H$ be the number of attention heads (SEAT is placed before attention; heads affect only optional per-head STG) and $G$ the number of spectral groups. We summarize time and parameter costs:

| Operation | FLOPs (dominant terms) | Params |
|---|---|---|
| rFFT | $\Theta(B\,d\,L\log L)$ | 0 |
| Real/Imag Linear (NIDP) | $\Theta(B\,L\,d^2)$ | $2d^2 + 2d$ |
| STG (Top-$k$ over $G$ groups) | $\Theta(H\,G\log G)$ | $H\,G$ |
| irFFT | $\Theta(B\,d\,L\log L)$ | 0 |
| Residual & Dropout | $\Theta(B\,L\,d)$ | $1\,(\alpha)$ |

Space: $\mathcal{O}(B\,L\,d)$ activations $+\ \mathcal{O}(d^2)$ weights. When $d$ is large, a low-rank factorization $W_\bullet = U_\bullet V_\bullet^\top$ with rank $r \ll d$ reduces both to $\Theta(B\,L\,dr)$ FLOPs and $2dr$ params.

## F   ANALYSIS OF ATTENTION STRUCTURE

To further investigate the effectiveness of the proposed SEAT module, we visualize the attention distribution before and after incorporating SEAT through heatmaps on the ETTh2 dataset. As shown in Figure 9, the baseline model exhibits block-like attention patterns that suggest limited expressiveness and an inability to differentiate between important and irrelevant temporal dependencies. These dense and repetitive structures indicate that the model tends to treat different time steps and features uniformly, which may lead to suboptimal forecasting. In contrast, after incorporating SEAT, the attention maps display significantly enhanced granularity and discriminative patterns, with sharper focus on salient regions. This improvement demonstrates SEAT's capacity to enrich the model's representational power by promoting structured, sparse, and information-aware attention.

Moreover, we further analyze the representational properties of the attention maps by evaluating two key metrics: attention entropy and the Top-5 Singular Value Energy Ratio (SVER), as shown in Figure 10. These metrics offer complementary perspectives on the sparsity, diversity, and expressive capacity of the learned attention patterns.

We introduce **attention entropy** $\mathcal{H}$ to quantify the uncertainty or dispersion in attention distributions. For a normalized attention matrix $\mathbf{M}$, it is defined as:

$$\mathcal{H}(\mathbf{M}) = -\sum_i \mathbf{M}_i \log \mathbf{M}_i, \tag{6}$$

where $\mathbf{M}_i$ denotes normalized attention weights. Lower entropy indicates more focused and interpretable attention, while higher entropy suggests a more diffuse distribution that may introduce noise. Attention entropy is closely related to the concept of matrix rank, particularly the effective rank, which is computed based on the entropy of normalized singular values:

$$\mathrm{eff\_rank}(\mathbf{M}) = \exp\left(-\sum_i p_i \log p_i\right), \quad \text{where } p_i = \frac{\sigma_i}{\sum_j \sigma_j}, \tag{7}$$

with $\sigma_i$ denoting the singular values of $M$. A skewed singular value distribution (i.e., a few dominant $\sigma_i$) results in lower entropy and thus a lower effective rank, indicating redundancy. Conversely, a higher effective rank suggests richer and more diverse representations. To further assess low-rankness, we compute the Top-5 Singular Value Energy Ratio (SVER):

$$\mathrm{SVER}_{\mathrm{Top\_5}} = \frac{\sum_{i=1}^{5} \sigma_i^2}{\sum_{j=1}^{n} \sigma_j^2}, \tag{8}$$

which measures the proportion of total matrix energy captured by the top-5 singular values. A higher SVER indicates that most of the information is concentrated in a few dominant components, reinforcing the interpretation of a low-rank structure.

**Discussion.**   The goal of introducing the SEAT module is to enhance the entropy of the attention distributions, thereby encouraging a more uniform and informative allocation of attention weights. This generally leads to a higher effective rank and a reduced dominance of top singular values, which indicates a less redundant, more expressive attention structure. From the results in Figure 10,

we observe that most models—such as PatchTST, Crossformer, and Informer—exhibit increased effective ranks and slightly reduced Top-5 Singular Value Energy Ratios after integrating SEAT. This confirms that SEAT helps to diversify attention representations and avoid over-concentration of energy in a few dominant components. Interestingly, iTransformer shows an exception: while its Top-5 SVER significantly increases from 0.891 to 0.986, its effective rank drops from 5.54 to 2.02. This suggests that, although SEAT introduces stronger regularization and reduces noise in attention, it may also lead to a more concentrated representation in already expressive models like iTransformer. This trade-off highlights that SEAT's impact may vary depending on the inherent structure and flexibility of the base model.

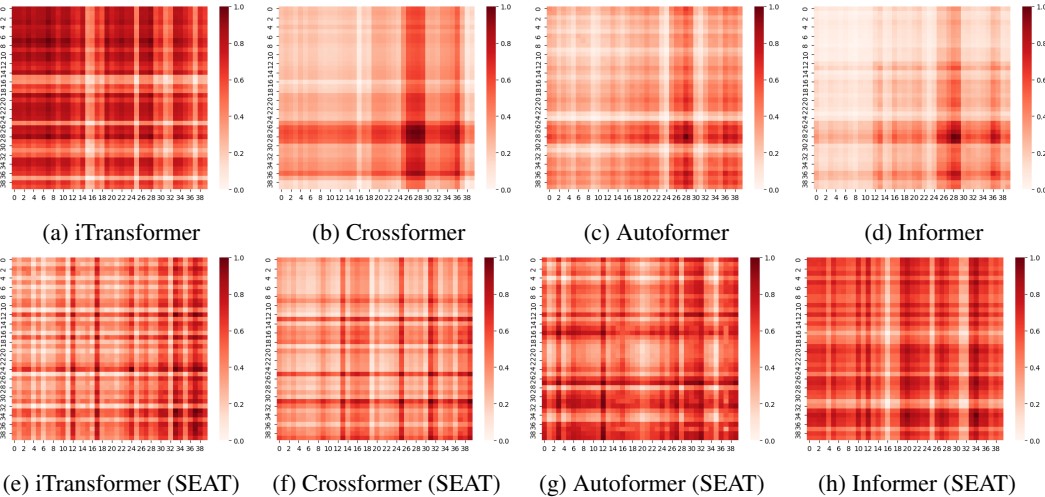

(a) iTransformer     (b) Crossformer     (c) Autoformer     (d) Informer

(e) iTransformer (SEAT)    (f) Crossformer (SEAT)    (g) Autoformer (SEAT)    (h) Informer (SEAT)

Figure 9: Attention heatmaps of the original model (a–d) and the model enhanced with SEAT (e–h) on the ETTh2 dataset. The original model exhibits obvious block-wise attention patterns with low discriminability. After introducing SEAT, the block structure is alleviated and the attention maps show improved focus and sharper distinctions, indicating better global awareness and feature separation.

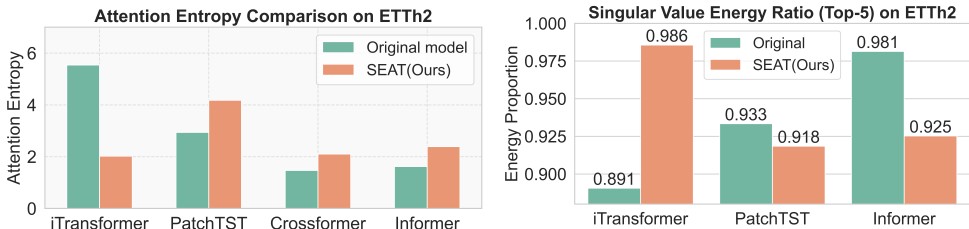

Figure 10: (a) Attention entropy between original models and their SEAT-augmented counterparts on ETTh2 dataset (Left). Lower entropy indicates less diverse attention, suggesting reduced representational capacity. SEAT enhances representational diversity by increasing attention entropy.(b) Comparison of the energy ratio captured by the top-5 singular values of attention matrices across different models on the ETTh2 dataset (Right). Models equipped with the SEAT (Ours) module exhibit lower energy concentration in the top singular values, indicating reduced low-rank bias and improved representational richness in the attention mechanism.

## G    VISUALIZATION OF PREDICTION RESULTS

To provide a clear comparison between models with and without SEAT, Figure 11, 12 and 13 illustrate the performance of three representative models: iTransformer, PatchTST, and Crossformer, both in their vanilla and SEAT-enhanced forms. All models benefit from SEAT integration, with iTransformer achieving the most accurate prediction of future series variations and demonstrating the best overall performance.

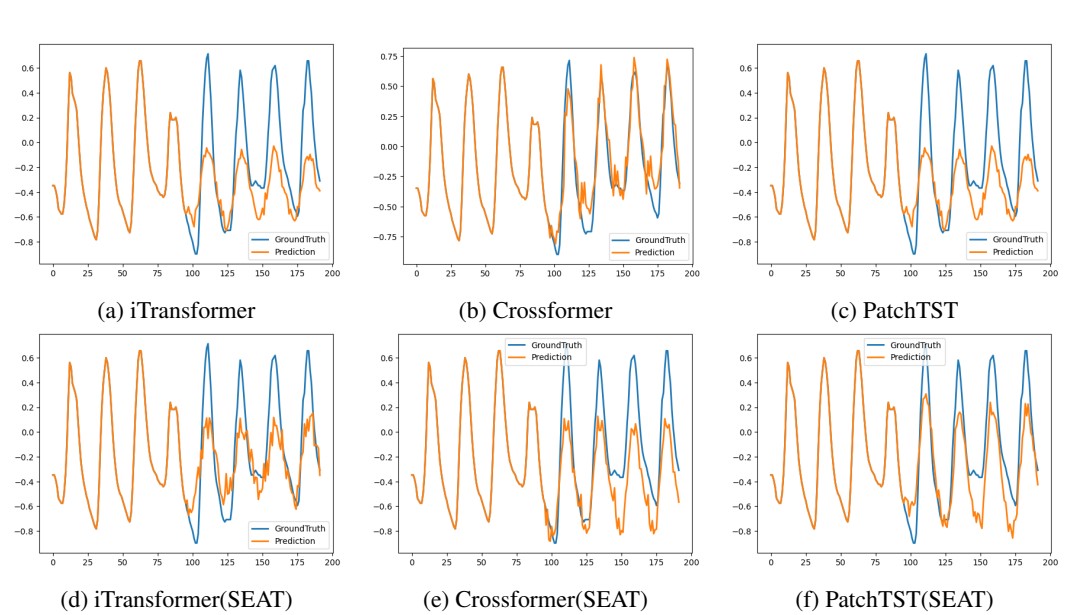

(a) iTransformer      (b) Crossformer      (c) PatchTST

(d) iTransformer(SEAT)      (e) Crossformer(SEAT)      (f) PatchTST(SEAT)

Figure 11: Visualization of input-96-predict-96 results on the ETTh2 dataset

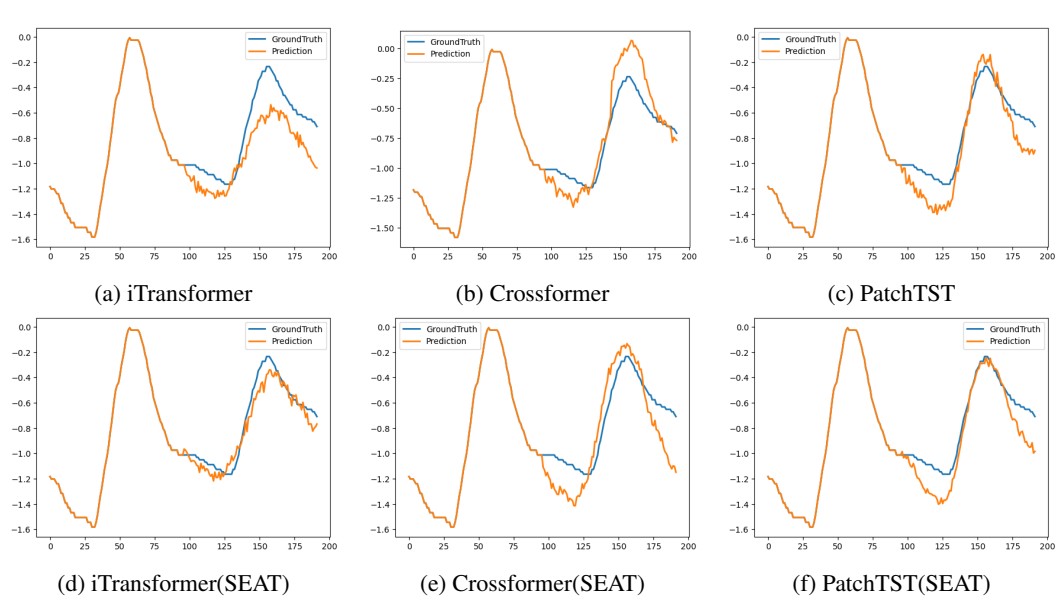

(a) iTransformer      (b) Crossformer      (c) PatchTST

(d) iTransformer(SEAT)      (e) Crossformer(SEAT)      (f) PatchTST(SEAT)

Figure 12: Visualization of input-96-predict-96 results on the ETTm2 dataset

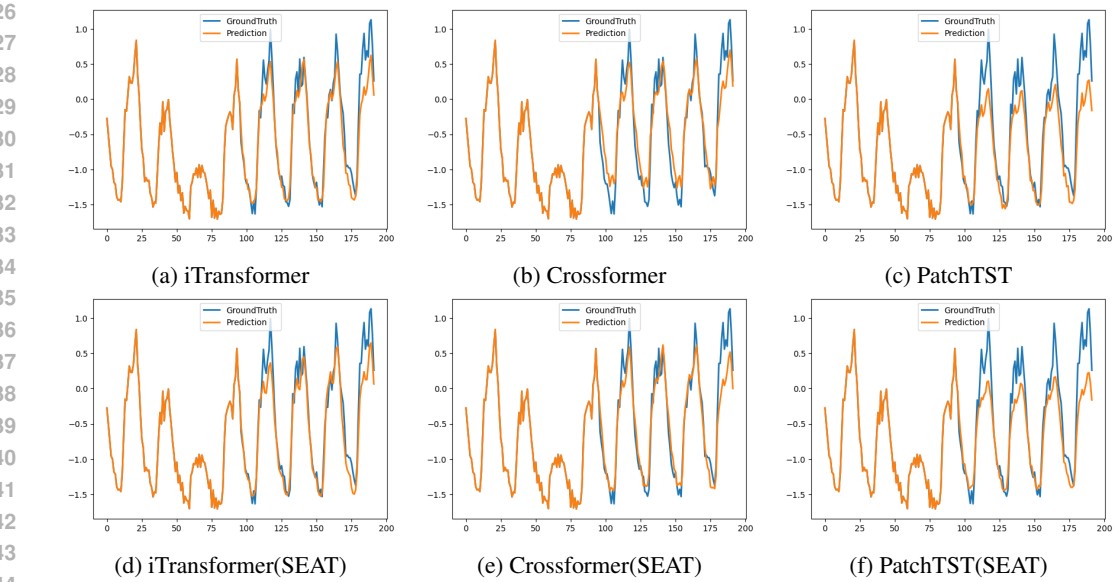

Figure 13: Visualization of input-96-predict-96 results on the ECL dataset

# H  BLOCK ATTENTION STRUCTURE: DERIVED PATTERNS AND MANIFESTATIONS

In this section, we provide theoretical and empirical insights into several structural properties observed in attention matrices of transformer-based time series forecasting models. These patterns, especially block-wise structures, are further analyzed through mathematical formulations and explanations.

**Diagonal Concentration and Band Patterns.**  The diagonally concentrated attention arises due to strong local temporal autocorrelation. Let $x_t$ be a time series such that $\mathbb{E}[(x_t - \mu)(x_{t+\tau} - \mu)] = \gamma(\tau)$ is the autocovariance function. For many real-world time series, $\gamma(\tau)$ decays rapidly as $|\tau|$ increases.

Given that attention weights are computed via:

$$A_{ij} = \frac{\exp(q_i^\top k_j)}{\sum_{j'} \exp(q_i^\top k_{j'})}$$

where $q_i, k_j$ depend on $x_i, x_j$, local autocorrelation implies $q_i^\top k_j$ is maximized when $|i - j|$ is small. Hence, attention concentrates along the diagonal.

**Periodic Block Replication.**  For time series with period $P$, i.e., $x_t \approx x_{t+P}$, this periodicity induces a block structure in the attention maps, repeating every $P$ steps. Formally, if $q_i = q_{i+P}$ and $k_j = k_{j+P}$ approximately hold, then:

$$A_{i,j} \approx A_{i+P,j+P}$$

This implies periodic blocks in the attention matrix. Fourier decomposition can be used to show that dominant frequencies in $x_t$ lead to structural repetitions in learned attention patterns.

**Asymmetric Block Structures.**  In causal attention, the attention mask enforces $A_{ij} = 0$ for $j > i$. This lower triangular structure leads to asymmetric blocks where attention is only allocated to the past:

$$A = \mathrm{softmax}(QK^\top + M), \quad M_{ij} = \begin{cases} 0, & j \leq i \\ -\infty, & j > i \end{cases}$$

The asymmetry is thus a direct consequence of this masking. Empirically, the asymmetry can be quantified via:

$$\mathrm{Asym}(A) = \|A - A^\top\|_F$$

which is significantly nonzero in such cases.

**Robustness to Noise and Overfitting.** Block attention can be interpreted as a form of regularization. Let $A = A^{\text{block}} + \epsilon$, where $A^{\text{block}}$ is the structured component and $\epsilon$ denotes noise or overfitting-induced irregularity.

A bounded Frobenius norm of $\epsilon$:

$$\|\epsilon\|_F^2 \ll \|A^{\text{block}}\|_F^2$$

implies model robustness. Furthermore, using a block mask or SEAT-like structure restricts attention to semantically meaningful groups, effectively reducing variance and thus generalization error according to the bias-variance tradeoff.

## I    RELATED WORKS

**Time-domain-based Transformers:** Transformer-based architectures have spurred extensive work on improving temporal learning for multivariate time series. For example, Autoformer (Wu et al., 2021) introduces an auto-correlation mechanism to capture inherent periodic patterns in time series data, while Pyraformer (Liu et al., 2022a) adopts a hierarchical attention structure to extract multi-scale temporal features. Crossformer (Zhang & Yan, 2023) utilizes a channel-dependent strategy to model cross-variate interactions. Other variants, such as the Nonstationary Transformer (Liu et al., 2022b) and iTransformer (Liu et al., 2024a), tackle challenges related to distributional shifts and complex multivariate dependencies, respectively. More recently, TwinsFormer (Zhou et al., 2025) introduces a dual-stream interaction mechanism that integrates trend and seasonal components via both the attention and feedforward layers, offering deeper alignment with the temporal dynamics.

**Frequency-domain-based Transformers:** Parallel to advances in the time domain, recent research has increasingly explored frequency-domain modeling as a compelling alternative. These methods aim to better capture global dependencies and periodic structures by transforming sequences into the spectral domain. Early examples such as FNet (Lee-Thorp et al., 2021) and AFNO (Guibas et al., 2021) leverage Fourier transforms to reduce computational complexity while maintaining strong predictive performance. FEDformer (Zhou et al., 2022) further combines spectral operations with self-attention to enhance long-range forecasting. FITS (Xu et al., 2024) and Fredformer (Piao et al., 2024) introduce learnable multi-band filters to address frequency bias. More recently, Kang et al. (Kang et al., 2024) propose spectral attention mechanisms to better capture long-period dependencies, and TIMEMIXER++ (Wang et al., 2025b) integrates multi-scale time-domain and multi-resolution frequency-domain features using hybrid strategies tailored for task-specific adaptation. FreDF (Wang et al., 2025a) offers a novel task projection approach in the frequency domain to reduce multi-step forecasting errors and enhance precision.

**Motivation and Positioning.** Despite recent progress, time-domain Transformers for long-horizon forecasting often suffer from block-wise attention collapse: strong local correlations dominate, yielding low-rank attention patterns that obscure global dependencies and inter-variable dynamics. On the other hand, frequency-domain approaches improve scalability and capture periodicity, yet they frequently disrupt the inherent smoothness and temporal coherence of raw signals, leading to structurally degraded attention maps and diminished feature expressiveness. These shortcomings are complementary but remain unresolved by existing methods. To bridge this gap, we propose **SEAT**, a lightweight and model-agnostic framework that spectrally conditions attention, provably raising its effective rank and entropy, thereby enabling richer and more reliable modeling of long-range and cross-variable dependencies.

## J    APPENDIX: DETAILED THEORETICAL JUSTIFICATION OF SEAT

**Notation.** For a window length $L$, let $\mathbf{F} \in \mathbb{C}^{L \times L}$ be the unitary DFT, $\mathbf{F}^*\mathbf{F} = \mathbf{I}$. For a WSS process with autocovariance $r(\tau)$ and spectral density $f(\omega)$, let $\Sigma$ be its $L \times L$ Toeplitz covariance. We use spectral norm $\|\cdot\|_2$, Frobenius norm $\|\cdot\|_F$, and stable rank $\text{sr}(\mathbf{A}) = \|\mathbf{A}\|_F^2 / \|\mathbf{A}\|_2^2$.

### J.1    SPECTRAL PRECONDITIONING AS APPROXIMATE WHITENING

**DFT diagonalizes circulant covariances.** Let $\mathbf{C}$ be the circulant matrix generated by $(r(0), r(1), \ldots, r(L-1))^\top$. Then $\mathbf{FCF}^* = \text{diag}(\lambda_k)$, where $\lambda_k$ sample the spectral density (up to scaling) (Gray, 2006). $\qquad\square$

**Toeplitz–circulant approximation.** If $\sum_\tau |r(\tau)| < \infty$, then there exists a circulant $\mathbf{C}$ with $\|\boldsymbol{\Sigma} - \mathbf{C}\|_2 \leq \eta_L$ and $\eta_L \to 0$ as $L \to \infty$ (Gray, 2006; Tilli, 1998). $\qquad\square$

**Approximate whitening.** Assume $0 < c \leq f(\omega) \leq C < \infty$. Let $\mathbf{D} = \mathrm{diag}(f(\omega_k)^{-1/2})$ and $\mathbf{M} = \mathbf{F}^*\mathbf{D}\mathbf{F}$. Then $\|\mathbf{M}\boldsymbol{\Sigma}\mathbf{M}^\top - \mathbf{I}\|_2 \leq c^{-1}\eta_L$. *Proof.* Decompose $\boldsymbol{\Sigma} = \mathbf{C} + (\boldsymbol{\Sigma} - \mathbf{C})$ and use Lemmas J.1–J.1. $\square$ $\qquad\square$

## J.2 NEAR-ISOMETRY AND EFFECTIVE RANK

Let $\mathbf{Z}$ stack $n$ windows; set $\widetilde{\mathbf{Z}} = \mathbf{Z}\mathbf{M}^\top$. Let $\mathbf{W}$ be an OSE/JL-type map with $m = \Theta(\varepsilon^{-2}\log n)$ that is $(1 \pm \varepsilon)$ on $\mathrm{span}(\widetilde{\mathbf{Z}})$ (Ailon & Chazelle, 2006; Ailon & Liberty, 2009; Sarlós, 2006; Clarkson & Woodruff, 2013).

**Stable-rank preservation.** If $\mathbf{P}$ is $(1 \pm \varepsilon)$ on the column space of $\mathbf{A}$, then $\mathrm{sr}(\mathbf{P}\mathbf{A}) \geq \frac{(1-\varepsilon)^2}{(1+\varepsilon)^2}\mathrm{sr}(\mathbf{A})$. $\qquad\square$

**Effective rank increase.** Under Theorem J.1 and the OSE above, with prob. $\geq 1 - \delta$, $\mathrm{sr}(\mathbf{W}\widetilde{\mathbf{Z}}) \geq \frac{(1-\varepsilon)^2}{(1+\varepsilon)^2}\mathrm{sr}(\widetilde{\mathbf{Z}})$. Moreover, by matrix concentration for (approx.) isotropic subgaussian rows (Vershynin, 2018; Tropp, 2015), $\mathrm{sr}(\widetilde{\mathbf{Z}}) \gtrsim \frac{d_{\mathrm{eff}}}{(1+c\sqrt{d_{\mathrm{eff}}/n})^2}$. $\qquad\square$

## J.3 ATTENTION ENTROPY AND GRADIENT DIVERSITY

Let $a_{ij} = \langle \mathbf{q}_i, \mathbf{k}_j \rangle / \sqrt{d_q}$ and $\mathbf{p}_i = \mathrm{softmax}(\mathbf{a}_i)$.

**Entropy via logit range.** Let $\Delta_i = \max_j a_{ij} - \min_j a_{ij}$ over $n$ keys. Then $H(\mathbf{p}_i) \geq \log n - \Delta_i$. *Proof.* Use $\sum_j e^{a_{ij}} \geq n e^{\min a}$ and $\sum_j p_{ij} a_{ij} \leq \max a$. $\square$ $\qquad\square$

**High-entropy rows after whitening+OSE.** If $(\mathbf{Q}, \mathbf{K})$ rows are (approx.) independent isotropic subgaussian vectors in $\mathbb{R}^{d_q}$, then with prob. $\geq 1 - 2/n$, $\Delta_i \leq C\sqrt{2\log n}$ and thus $H(\mathbf{p}_i) \geq \log n - C\sqrt{2\log n}$ (Boucheron et al., 2013). $\qquad\square$

**Gradient alignment bound.** In the small-logit regime ($|a_{ij}| \leq \alpha$), the gradient contributions $\mathbf{G}_i$ to $\mathbf{W}_Q$ satisfy $\mathbb{E}[\cos(\mathbf{G}_i, \mathbf{G}_j)] \leq C_1 \mathrm{coh}(\mathbf{Q}) + C_2\alpha + o(1)$, where $\mathrm{coh}(\mathbf{Q}) = \max_{u \neq v} \frac{|\langle \mathbf{q}_u, \mathbf{q}_v \rangle|}{\|\mathbf{q}_u\|_2\|\mathbf{q}_v\|_2}$. *Sketch.* Linearize softmax around uniform, backpropagate through $\mathbf{a}_i = \mathbf{Q}\mathbf{K}^\top/\sqrt{d_q}$, and take expectations. $\qquad\square$

## J.4 GENERALIZATION VIA SPECTRAL GROUP SPARSITY

Partition the frequency axis into $G$ bands; let $\mathbf{D}$ activate $s \ll G$ of them. For a linear head with group-lasso norm $\leq B$, the Rademacher complexity of the induced class on SEAT features satisfies

**Group-sparse complexity.** $\mathfrak{R}_n \lesssim \frac{B}{\sqrt{n}}\sqrt{s\log(1 + G/s)}$ up to constants depending on subgaussian parameters (Maurer & Pontil, 2012; Lounici et al., 2011). $\qquad\square$

## J.5 EXTENSIONS: CROSS-SPECTRAL MIXING AND CAUSALITY

A light cross-channel mixer $\mathbf{M}_\times(\omega_k)$ per frequency preserves near-isometry if near-orthogonal (block-orthogonal extension of Theorem J.2). Causal STFT ensures non-anticipativity: each window ends at the prediction time, preventing information leakage.