# OpenReview forum: "SEAT: Sparsity Enhanced Attention in the frequency domain for Time Series forecasting"
_ICLR.cc/2026/Conference — Submitted to ICLR 2026_

### Official Review · Reviewer_EASt · 2025-10-30

**Soundness:** 2
**Presentation:** 2
**Contribution:** 2
**Rating:** 2
**Confidence:** 5

**Summary:**

This paper proposes SEAT (Sparsity-Enhanced Attention in the Frequency Domain), a lightweight spectral module designed to mitigate the block-wise redundancy problem in Transformer-based time-series forecasting models. The authors argue that strong local smoothness and inter-variable correlation lead to redundant low-rank attention maps and oversmoothed representations. Experiments on multiple forecasting benchmarks show consistent performance gains, and the authors also provide some theoretical analyses related to rank preservation and generalization.

**Strengths:**

The paper is well-written and easy to follow.

The proposed module is lightweight and can be easily applied to existing Transformer forecasters.

Empirical results show moderate yet consistent improvements across several benchmarks.

**Weaknesses:**

The core idea of this paper is to address redundancy or smoothness through spectral preconditioning. This closely resembles previous studies such as Fredformer (KDD’24),  FilterNet (NeurIPS’24), and FredDF (MM24), which all explore frequency-domain reweighting or decorrelation for time-series forecasting. This paper claim of tackling *block-wise attention collapse* is essentially a restatement of the same frequency-imbalance / oversmoothing issue under different terminology. The paper does not clearly articulate what new insight is brought beyond these existing works, nor how SEAT differs conceptually from recent frequency-domain modeling methods. In general, the motivation is unclear as well.

The proposed module combines standard FFT preprocessing with orthogonal linear projections and optional Top-K selection. These components are individually well-established and their integration appears heuristic rather than theoretically grounded. The “near-isometric dual projection” essentially performs orthogonal linear mixing, which resembles decorrelation layers used in earlier spectral normalization. Sometimes Top-K is an empirical way that cannot ensure the selection is always satisfied and easily influenced by noise. How to evaluate its effectiveness and necessity?

While SEAT improves multiple baselines, most comparisons are made only against common time-domain models (e.g., iTransformer, PatchTST) in Table 1. There is lacking of frequency modeling baselines (Fredformer, FreDF, etc.) makes it unclear whether SEAT is truly better or simply another reweighted frequency regularization. The improvements shown in Table 2 are somewhat expected, as adding an additional frequency module to time-domain models naturally complements missing spectral representations; therefore, the observed gains primarily demonstrate incremental complementarity, rather than a fundamentally new modeling capability.

Moreover, no analysis is provided on computational overhead, energy spectrum changes, or ablations comparing different forms of orthogonalization, as such it is difficult to attribute improvements to the proposed mechanisms.

Overall, there are incremental contribution in this paper. The community has recently seen sevearl works tackling time-series attention degeneration through spectral or low-rank perspectives. The paper presents a neat combination of known tricks rather than a principled method or analysis.


-------

**Minor Issues**

- Some notations and proofs are unnecessarily long.
- The Top-K spectral gating is only briefly discussed and lacks interpretability experiments.
- Fig 2 and 3 visualize attention patterns but could include quantitative diversity metrics.

**Questions:**

Please kindly refer to the weaknesses

---

### Official Review · Reviewer_BbR7 · 2025-11-01

**Soundness:** 2
**Presentation:** 2
**Contribution:** 3
**Rating:** 4
**Confidence:** 3

**Summary:**

This manuscript identifies and formalizes a block-wise attention collapse phenomenon in time-series Transformers. It proposes Sparsity-Enhanced Attention in the Frequency Domain, a lightweight, plug-and-play pre-attention module. The paper analyzes SEAT with theoretical properties and claims O(L log L) overhead with minimal parameters.

**Strengths:**

The block-wise collapse is articulated with entropy-rank diagnostics and gradient-alignment arguments, offering a persuasive motivation for spectral preconditioning and rank enhancement.

SEAT slots in before the attention block, adds little params, and preserves the backbone kernel. As a result, it can be easily applied to existing transformer-based methods.

Experimental results cover eight datasets and multiple baselines. The plug-in experiments show consistent improvements.

**Weaknesses:**

The SEAT block first transforms the time series into the frequency domain and then applies an inverse transform, but the manuscript does not explore whether frequency-domain features could be directly integrated with the attention mechanism.

The number of learnable parameters in SEAT is relatively small. It is unclear whether stacking SEAT and Transformer blocks would further improve representation capacity and accuracy.

The current design does not leverage the patching mechanism popularized by PatchTST, which is effective for time-series forecasting; integrating patch-level processing might yield additional gains.

**Questions:**

(a) Please experiment with applying the Transformer directly in the frequency domain and report the results to validate whether frequency-aware attention provides benefits.

(b) Consider stacking multiple SEAT + Transformer blocks and analyze performance and efficiency trade-offs.

(c) Include the patch mechanism so that SEAT and the Transformer can operate at the patch level, and compare against the current setting.

---

### Official Review · Reviewer_cFDf · 2025-11-01

**Soundness:** 2
**Presentation:** 2
**Contribution:** 2
**Rating:** 4
**Confidence:** 3

**Summary:**

This paper addresses a structural weakness in transformer-based time series forecasting models: the tendency of self-attention to collapse into block-wise patterns, where adjacent time steps receive nearly identical attention weights. This low-rank behavior restricts representational diversity and hampers long-range temporal modeling. The authors theoretically characterize this “block-wise attention collapse” by linking it to temporal autocorrelation and cross-variable redundancy, showing that such correlations reduce the effective rank of attention matrices and align gradients, leading to optimization stagnation.

To mitigate this, they propose SEAT (Sparsity-Enhanced Attention in the Frequency Domain), a lightweight, model-agnostic module inserted before the attention block. SEAT performs FFT-based spectral preconditioning, near-isometric linear projection on real and imaginary components, and optional top-k frequency gating before reconstructing the signal via inverse FFT. This process whitens temporal statistics, disentangles cross-variable correlations, and increases attention entropy and rank. The authors provide formal guarantees on approximate whitening, stable-rank preservation, and gradient diversity. Empirically, SEAT consistently outperforms baselines across eight standard multivariate forecasting datasets (ETT, Weather, ECL, Exchange, Traffic), improving MSE and MAE by 5–80% depending on the backbone. Ablations show both the frequency transform and residual fusion are essential, and visualization confirms SEAT suppresses block-like patterns while broadening the attention spectrum.

**Strengths:**

First, it establishes a solid theoretical foundation, rigorously analyzing how temporal smoothness and cross-variable dependencies induce low-rank, block-wise attention collapse and gradient alignment, thus framing a precise expressivity limitation of transformer attention.
Second, it proposes an elegant and general spectral solution (the SEAT module) which leverages FFT-based whitening and near-isometric projection to decorrelate signals while remaining fully plug-and-play and computationally lightweight. Third, its empirical validation is extensive and convincing, spanning eight real-world benchmarks and seven transformer architectures, consistently yielding state-of-the-art accuracy improvements with negligible cost. Finally, the work delivers clear interpretability and diagnostic clarity, showing through entropy and rank analyses that SEAT restores attention diversity and global mixing, thereby unifying theoretical prediction and empirical evidence into a coherent framework for future transformer design.

**Weaknesses:**

1. The theoretical guarantees strongly depend on assumptions such as wide-sense stationarity (WSS), isotropic subgaussian conditions, and near-isometric projection. These assumptions rarely hold true in realistic multivariate time-series forecasting scenarios, limiting the practical applicability of the presented theoretical results.

2. Empirical evaluations indicate inconsistent results across different architectures (notably, Informer) and datasets (e.g., Traffic). Additionally, evaluations primarily use a fixed sequence length of 96, potentially masking issues arising at longer forecast horizons or high-dimensional feature spaces.

3. The roles and interactions among FFT-based spectral transforms, RevIN normalization, and near-isometric dual projections are not sufficiently isolated. Furthermore, the causality of the FFT–inverse FFT procedure, crucial for sequential forecasting scenarios, is not explicitly validated or discussed in depth.

**Questions:**

1. Can the authors empirically demonstrate how deviations from the theoretical assumptions (WSS, isotropy, near-isometry) quantitatively affect the link between improved attention entropy/rank and forecasting accuracy?

2/ Could the authors provide extensive evaluations on significantly longer forecasting horizons (e.g., ≥1,440 steps) and higher-dimensional feature spaces, and explain the observed performance degradation on architectures like Informer?

3. Could the authors separately quantify the contributions of spectral transforms, RevIN normalization, and orthogonality regularization? Moreover, how do the authors ensure strict causality during the FFT–inverse FFT preprocessing stage for rolling-window forecasts?

---

### Official Review · Reviewer_ay8m · 2025-11-03

**Soundness:** 1
**Presentation:** 2
**Contribution:** 2
**Rating:** 2
**Confidence:** 4

**Summary:**

This paper proposes Sparsity-Enhanced Attention in the Frequency Domain (SEAT), a lightweight and plug-and-play module designed to mitigate block-wise patterns in the attention matrix by preprocessing time-series data in the frequency domain. This approach disentangles local smoothness from cross-variable redundancy, leading to more diverse attention patterns. The method is supported by theoretical analyses and empirical evidence, and integrating SEAT into various forecasting architectures consistently improves their performance across multiple benchmark datasets, achieving state-of-the-art results.

**Strengths:**

- The proposed SEAT module is simple and can be easily integrated into various architectures.
- Incorporating SEAT into different architectures improves their forecasting performance.

**Weaknesses:**

Overall, The contributions are not sufficiently supported.
- Unconvincing theoretical foundation.
  - It is unclear how the provided theorems are related to the proposed method. The theoretical analyses are too brief, and there is no detailed explanation of how each statement in Section J justifies the design of the method. Since theoretical foundation is a main contribution of this paper, that should be clearly described.
  - The condition described by Eq (1) does not imply that the rank of the attention matrix is low. Eq (1) only implies that $\mathbf{A}_{t,s} \approx 0$ for $t \in \mathcal{C}$ and $s \notin \mathcal{C}$. For instance, if Eq (1) holds and $\text{sim}(\mathbf{q}_i, \mathbf{k}_i) \gg \text{sim}(\mathbf{q}_i, \mathbf{k}_j)$ for all $i, j \in \mathcal{C}$ and $i \neq j$, then the attention matrix has rank $\ge m$.
  - Similarly, Eq (3) does not imply that the rank of the attention matrix is low.
  - The SEAT block preprocesses input time series in the frequency domain before the main encoder (i.e., feature extractor), which employs attention layers in the temporal domain. How SEAT affects the temporal attention behavior is unclear.
  - Recent architectures rely on temporal patches (or segments). It is not clear whether the theoretical analyses in the paper still hold under such patch-based designs.
- Weak empirical evidence.
  - Figure 2 only shows Attention Entropy, not effective rank. Moreover, the figure does not define what range (e.g., 2-4) means a "low" entropy value. A comparison with SEAT should be included in the figure. In addition, as shown in Figure 6, SEAT increases the attention entropy only marginally, which raises doubts about whether SEAT truly enables the model to capture more diverse attention patterns. Furthermore, Appendix F shows that iTransformer has even higher entropy than SEAT, contradicting the authors' claim.
  - The effective rank is never reported, despite being a key concept throughout the paper. The authors use attention entropy and effective rank interchangeably, but by definition they are distinct metrics. This distinction should be clarified.
  - The paper only reports SVER (top-5), which is indirectly related to effective rank. This provides only partial information. Please include more complete results, such as singular value distributions and effective rank computed via Eq (7) rather than top-5 SVER defined in Eq (8).
  - Could you also report attention entropy and effective rank across all datasets? As shown in Figure 6 (ETTm2) and Figure 10 (ETTh2), the trends may differ by dataset. To substantiate the claims, more comprehensive experiments are necessary.
- Insufficient empirical validation.
  - Several recent and strong baselines are missing, such as CARD [1], DeformableTST [2], and CATS [3]. These methods reportedly outperform or perform comparably to SEAT.
  - The experiments are conducted only with an input length of 96. Prior works (e.g., [2]) typically evaluate multiple input lengths (e.g., 384, 768), which should also be considered for a fair comparison.

[1] CARD: Channel aligned robust blend transformer for time series forecasting, ICLR 2024 \
[2] DeformableTST: Transformer for time series forecasting without over-reliance on patching, NeurIPS 2024 \
[3] Are self-attentions effective for time series forecasting?, NeurIPS 2024

**Questions:**

See the weaknesses part.

---

### Meta-Review · Area_Chair_VJ6W · 2026-01-05

**Summary:**

After careful consideration of all reviews, I recommend rejection of this submission. While the paper tackles an interesting problem of block-wise attention collapse in time series transformers, the reviewers have identified several fundamental issues that cannot be overlooked. The theoretical analysis, which the authors position as a main contribution, suffers from overly restrictive assumptions (WSS, isotropic subgaussian conditions) that rarely hold in real forecasting scenarios, and the connection between the theoretical results and the proposed method remains unclear. More critically, Reviewer EASt raises a compelling concern about novelty: the core idea of using frequency-domain preprocessing to address attention redundancy closely parallels recent works like Fredformer, FilterNet, and FredDF, yet the paper fails to differentiate itself conceptually or include these methods as baselines. The empirical validation is also problematic, testing only a single input length of 96 when recent works evaluate multiple horizons, omitting strong recent baselines like CARD and DeformableTST, and showing inconsistent improvements across architectures. The reviews collectively suggest this is an incremental combination of existing techniques rather than a principled new approach, and the authors need to substantially strengthen both the theoretical justification and experimental rigor to meet the conference standard.

**Reviewer Concerns:**

The lack of rebuttal means we have no evidence that authors can satisfactorily address the reviewers' concerns about novelty, theoretical rigor, and experimental comprehensiveness.

**Reviewer Scores:**

The lack of rebuttal means we have no evidence that the reviewers would change their scores.

---

### Decision · Program_Chairs · 2026-01-26

Reject